# Multifunctional non-woven fabrics of interfused graphene fibres

Zheng Li[1], Zhen Xu[1], Yingjun Liu[1], Ran Wang[1] & Chao Gao[1]

Carbon-based fibres hold promise for preparing multifunctional fabrics with electrical conductivity, thermal conductivity, permeability, flexibility and lightweight. However, these fabrics are of limited performance mainly because of the weak interaction between fibres. Here we report non-woven graphene fibre fabrics composed of randomly oriented and interfused graphene fibres with strong interfibre bonding. The all-graphene fabrics obtained through a wet-fusing assembly approach are porous and lightweight, showing high in-plane electrical conductivity up to $\sim 2.8 \times 10^4\,\mathrm{S\,m^{-1}}$ and prominent thermal conductivity of $\sim 301.5\,\mathrm{W\,m^{-1}\,K^{-1}}$. Given the low density ($0.22\,\mathrm{g\,cm^{-3}}$), their specific electrical and thermal conductivities set new records for carbon-based papers/fabrics and even surpass those of individual graphene fibres. The as-prepared fabrics are further used as ultrafast responding electrothermal heaters and durable oil-adsorbing felts, demonstrating their great potential as high-performance and multifunctional fabrics in real-world applications.

[1] MOE Key Laboratory of Macromolecular Synthesis and Functionalization, Department of Polymer Science and Engineering, Key Laboratory of Adsorption and Separation Materials & Technologies of Zhejiang Province, Zhejiang University, 38 Zheda Road, Hangzhou 310027, China. Correspondence and requests for materials should be addressed to C.G. (email: chaogao@zju.edu.cn).

Graphene fibres, assembled from graphene sheets, are expected to bring the ideal attributes of monolayer graphene into 1D fibrous materials in the macroscopic scale, and thus should grow as one of the most attractive new-style carbon-based fibres[1–4]. Nowadays, after several years of continuous effort, massive and economic production of graphene fibres is available[5,6], while their highest strength and modulus have already reached 2.2 and 400 GPa, respectively. Notably, the electrical and thermal conductivities up to $8 \times 10^5$ S m$^{-1}$ and 1,290 W m$^{-1}$ K$^{-1}$ are at the forefront of current carbon-based fibres[7,8]. Meanwhile, the commonly used flexible wet-assembly technique allows facile structural design and integration of functionalities on the as-prepared graphene fibres, whose merits have been disclosed in applications encompassing energy storage devices[9,10] and environmentally responsive systems[11–14].

The advanced applications of fibrous materials are mainly referred to making fabrics. In this regard, carbon-based fabrics are popular in many fields, such as energy, automobile, aerospace and so on because of their advantages in combining electrical and thermal conductivities, flame and chemical resistance, permeability and lightweight. However, there is a general problem in carbon-based fabrics that the realization of strong interfibre interaction remains challenging because of the refractory and insoluble nature of carbon materials. In consequence, the fabric performance is significantly degraded as compared with its constituent individual fibre, especially for the electrical and thermal conductivities, which are sensitive to interfacial resistance. Among the list of new-type carbon-based fabrics, graphene fibre fabrics (GFFs) should be extremely promising, so as to bring out the full potentials of such appealing fibres into a large scale. Some efforts have been made to prepare graphene fibre networks via chemical vapor deposition (CVD) or direct spinning methods[15–17], whereas the real GFFs with sufficient packing density of well-connected fibres to deliver high electrical and thermal conductivities have never been reported, owing to the lack of efficient fabrication approach.

In this work, we propose a scalable strategy to produce randomly laid non-woven GFFs constructed by graphene staple fibres (short-length fibres as distinct from continuous filaments). Through wet-fusing assembly of graphene oxide (GO) fibres in aqueous solvents, the independent fibres are integrated into a whole fabric with strong interfibre bonding. After being annealed at 3,000 °C, the interfused GFFs are found tough, flexible, lightweight and highly conductive. Their specific electrical and thermal conductivities are several times higher than those of former carbon-based papers/fabrics, even individual graphene fibres. We further demonstrate that our multifunctional GFFs perform well in application as electrothermal heaters with quick response. Besides, the GFFs show efficient adsorption for viscous organic solvents and oils, together with a good recyclability for more than 20 cycles.

## Results

**Fabrication of GFFs.** Radom deposition of staple fibres is widely used in the industry to produce non-woven fabrics of carbon fibres, polymeric fibres and so on. The non-woven configuration is convenient to reach an adequate packing density, thus facilitating the integral fabric performance, and is technically viable for industrial production of GFFs. Graphene fibres were prepared using a wet-spinning protocol[1,18–20]. To realize interfibre bonding, we initiate a wet-fusing assembly approach based on self-assembly of the as-prepared GO fibres. Three major issues should be resolved during the fabrication of non-woven GFFs: first, continuous spinning of GO staple fibres that build the framework; second, preventing severe shrinkage during drying to sustain the fabric structure; and third, achieving strong interaction between fibres to form a robust fibre network.

The detailed fabrication process is described in Fig. 1, including the following three main steps: spinning and drying of GO staple fibres (Fig. 1a,b), wet-fusing assembly into fabrics (Fig. 1c–e) and high-temperature annealing/reduction (Fig. 1f).

First, the continuous spinning of GO staple fibres was performed by injecting GO/DMF spinning dopes (5 mg ml$^{-1}$) into an ethyl acetate coagulation bath with rotation speed of 40–50 r.p.m. The rotating coagulation bath put excessive stretching on the as-extruded fibres via friction force and resulted in GO staple fibres with specific length. The staple length was easily controlled through adjusting the speed ratio of injection to rotation (Supplementary Fig. 1), as it increases with the speed ratio until a continuous fibre is obtained. While the position of the spinning nozzle was fixed, GO staple fibres with uniform length were obtained successively (Supplementary Movie 1). Then, the as-spun GO fibres were collected by filtration and dried below 60 °C. After re-dispersing of the dried fibres in the mixture of H$_2$O and ethanol (volume ratio of 3:1), a temporarily homogeneous GO fibre pulp was ready for fabric formation (Supplementary Fig. 2a). Subsequent filtration and drying gave

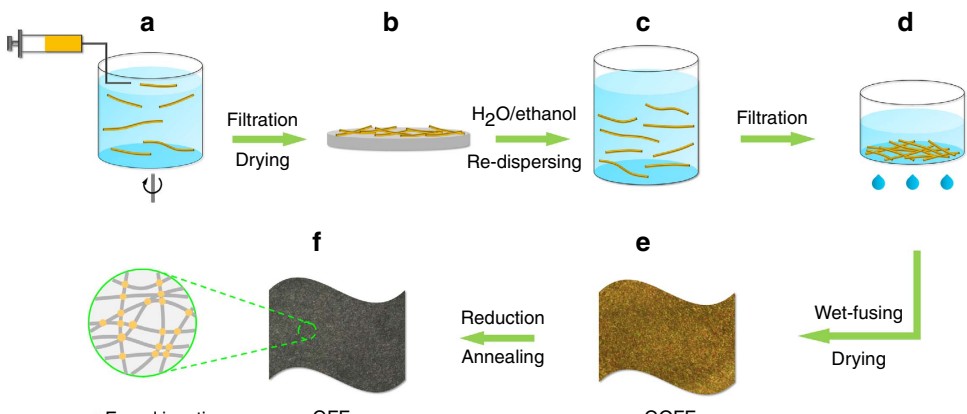

**Figure 1 | Fabrication of GFFs via wet-fusing assembly.** (**a**) Continuous wet-spinning of GO staple fibres. (**b**) First drying of the as-spun GO fibres. (**c**) Re-dispersion of dried GO fibres in the mixture of H$_2$O and ethanol. (**d**) Wet-fusing assembly of GO fibres after filtration of the re-dispersed fibres. (**e**) A free-standing GOFF with brownish colour after drying. (**f**) A grey GFF after chemical reduction or thermal annealing with randomly oriented and interfused graphene fibres.

rise to interfused GO fibre fabrics (GOFFs), which were further converted to GFFs by thermal annealing at 1,000 °C (GFF-1,000), 2,000 °C (GFF-2,000) and 3,000 °C (GFF-3,000), respectively. Chemical reduction using hydrazine hydrate ($N_2H_4 \cdot H_2O$) was also performed as a comparison (GFF-$N_2H_4$).

Notably, the above re-dispersion strategy is crucial for achieving well-defined GFFs. As shown in Supplementary Fig. 2a, first drying of GOFFs caused more than 90% of volume shrinkage to form highly compact irregular fibre stacks. The following re-dispersion process dispersed the aggregated fibres, re-assembled the separated fibres and facilitated the formation of regular net-like GOFFs. Meanwhile, the re-dispersed fibres only experienced slight contraction during drying (Supplementary Fig. 2b–d), allowing sustention of the established fabric structure.

**Mechanism of wet-fusing assembly**. The interfibre bonding was generated via wet-fusing assembly of the re-dispersed GO fibres. The assembly process was *in situ* traced by optical microscopy (OM). During re-dispersion in aqueous solvent, the dried GO fibres immediately wetted, gradually swelled and then turned into gel fibres through solvation[21,22]. The swollen gel fibres showed Schlieren texture under polarized-light OM because of the liquid crystal birefringence (Fig. 2a,b). The gel state of fibres is critical to realize strong interfibre interaction since the gel fibres can fuse at their contacts during subsequent drying. Figure 2c shows the assembly process of two crossed gel fibres. With the evaporation of solvent via natural drying, the two gel fibres got closer until they contacted with each other, and then quickly fused together to become one crossing fibre (Supplementary Movies 2 and 3). We coin this process as the wet-fusing assembly, which is contributed from the swelling of fibres and hydrogen bonding between GO

sheets across the interface. On the basis of such a mechanism, plenty of re-dispersed GO fibres interfused at cross points into an integrated GOFF.

**Morphology of GOFFs and GFFs**. Different from compact graphene papers made by the classic vacuum-assisted filtration[23], the GOFFs are porous and thus highly permeable to visible light. While keeping regular shape, the GOFFs were readily prepared with tailored thicknesses varying from tens of microns to several millimetres (Fig. 2d,e). The high-temperature annealing did not change the porous structure, which allowed penetration of light (Fig. 2f) and air (permeability of $2.14 \times 10^{12}\,cm^3\,m^{-2}\,h^{-1}$ at 0.1 MPa), but changed the colour from brown to deep grey because of the reduction of GO. Meanwhile, the contraction of fibres during annealing decreased the lateral dimension (7.7%) and thickness (53.9%) of GFFs (Fig. 2g). The resulting GFFs are so mechanically strong and flexible that they could be tailored into strips and coiled around a glass rod (Fig. 2h). Furthermore, based on the continuous spinning of GO staple fibres, the controllable and scalable fabrication of GFFs is easy to implement (Fig. 2i).

**Characterization of GOFFs and GFFs**. The as-prepared GOFFs contain a large amount of oxygen-containing functional groups on GO sheets with a C/O ratio of ~2.17, as characterized with X-ray photoelectron spectroscopy (XPS). After chemical reduction by $N_2H_4$, the functional groups were partially removed, and the GFF-$N_2H_4$ showed a decreased O1s peak in XPS spectrum and an increased C/O ratio ~7.42. Upon thermal annealing, the C/O ratio rose significantly as the annealing temperature increased from 1,000 to 3,000 °C. The O1s peak became indistinct

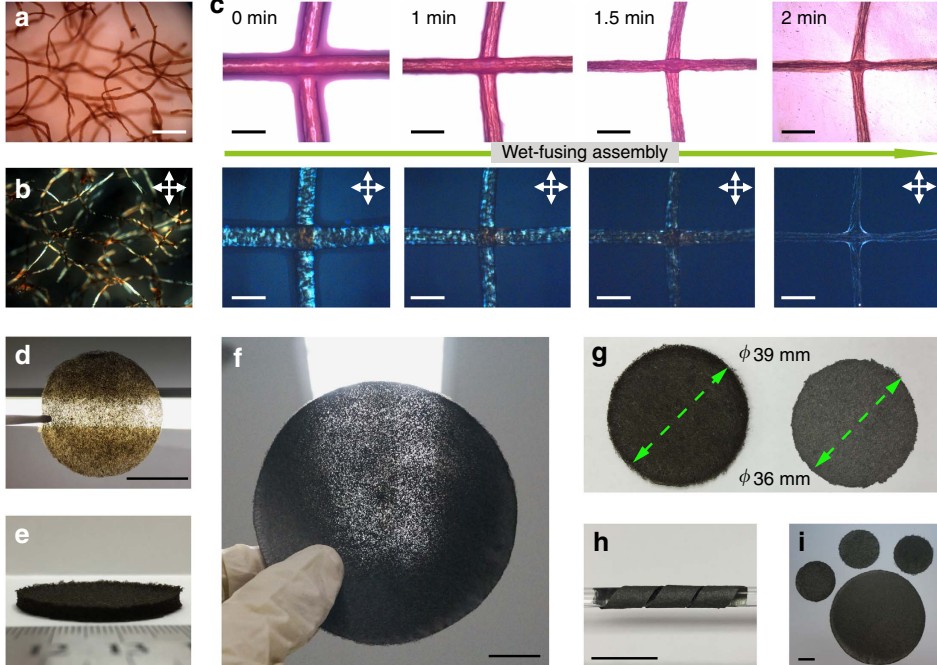

**Figure 2 | Mechanism of wet-fusing assembly and morphology of the as-prepared GOFFs and GFFs.** Micrographs of the re-dispersed GO fibres in a $H_2O$/ethanol mixture under (**a**) OM and (**b**) polarized-light optical microscopy (POM). (**c**) Wet-fusing of GO fibres recorded under OM and POM. Photographs of (**d**) a piece of thin GOFF (thickness 0.05 mm) held towards an light-emitting diode lamp, showing its porous structure and light brown colour, (**e**) a thick and dark brown GOFF (thickness 3 mm), (**f**) a thermally annealed GFF with porous feature for light and gas penetration, (**g**) GOFF (left) and GFF (right), indicating the slight shrinkage of lateral dimension and colour change, (**h**) a strip of GFF coiled around a glass rod and (**i**) four GFFs of different sizes and thicknesses. Scale bars, 500 μm (**a,b**), 150 μm (**c**) and 20 mm (**d,f,h,i**).

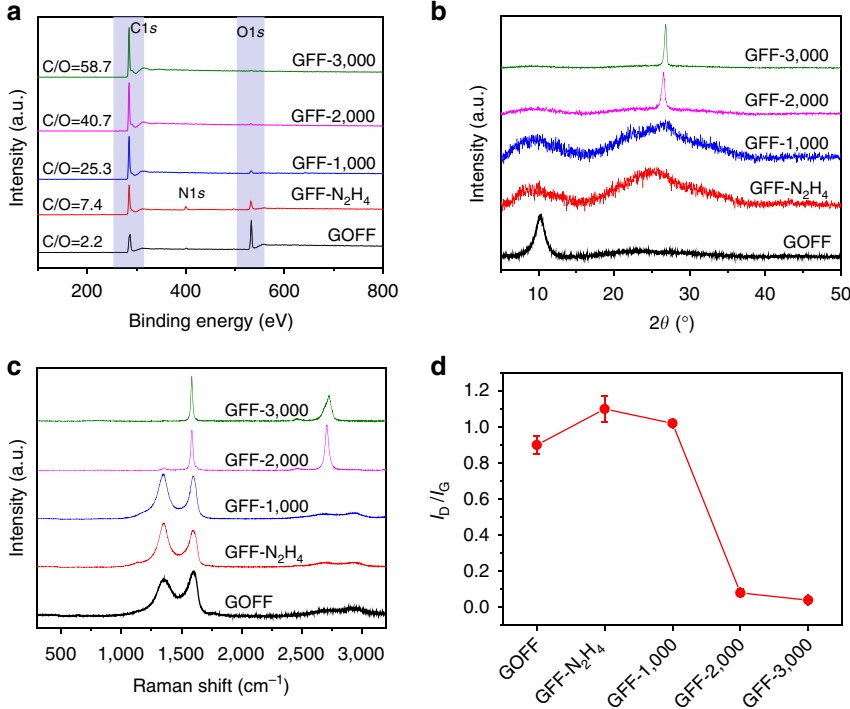

**Figure 3 | Characterization of GOFFs and GFFs.** (**a**) XPS spectra of the as-prepared GOFF, $N_2H_4$ reduced GFF and thermally annealed GFFs at 1,000, 2,000 and 3,000 °C. (**b**) X-ray diffraction patterns of GOFF, GFF-$N_2H_4$ and annealed GFFs. (**c**) Raman spectra of GOFF, GFF-$N_2H_4$ and annealed GFFs. (**d**) Variation of $I_D/I_G$ in different samples. Error bars represent the s.d. of $I_D/I_G$ for at least five measurements.

when the annealing temperature reached 2,000 °C or above, indicating the complete removal of functional groups (Fig. 3a and Supplementary Table 1). The changing is confirmed using X-ray diffraction analyses. The shift of X-ray diffraction peaks from 10.2° of GOFF to 26.5° of GFF-3,000 implies a decrease in interlayer spacing from 8.6 to 3.3 Å, as well as a high degree of graphitization of GFFs[24,25] (Fig. 3b). Raman spectra further reveal the structural evolution (Fig. 3c). The D to G peak intensity ratio ($I_D/I_G$) increased in GFF-$N_2H_4$ because the $N_2H_4$ reduction induced plenty of defects after partial removal of functional groups[26]. In contrast, thermal annealing led to continuous decrease in the $I_D/I_G$ ratio with increasing temperature, suggesting the effective healing of defects on graphene sheets (Fig. 3d). The narrowed G band and recovered 2D band since 2,000 °C annealing further prove the recovery of crystalline domain on graphene sheets at high temperatures[8,25]. In addition, the occurrence of asymmetric 2D band for GFF-3,000 is consistent with previous reports[8], revealing the presence of AB stacking of Bernal phase between graphene sheets[27]. Overall, the high-quality graphene with defect-free structure has been achieved in GFFs annealed at 3,000 °C.

**Electrical and thermal conductivities of GFFs.** The electrical and thermal conductivities of differently reduced GFFs were investigated. Two types of GFFs, 130GFFs and 200GFFs referring to fabrics resulting from a 130 or 200 μm spinneret at the very beginning, were tested in the measurement. As depicted in Fig. 4a, high-temperature annealing is favourable for conductive properties than chemical reduction. Both electrical and thermal conductivities are progressively improved with ascending temperature due to reduced defects and improved crystallinity of graphene sheets. The 130GFFs generally show better conductivities than 200GFFs in all cases of reduction protocol. Therefore, the best in-plane electrical and thermal conductivities were both

attained in 130GFF-3,000, with values of $\sim 2.8 \times 10^4 \, \mathrm{S \, m^{-1}}$ and $301.5 \, \mathrm{W \, m^{-1} \, K^{-1}}$, respectively. Significantly, the in-plane thermal conductivity of porous 130GFF-3,000 exceeds those of aluminium ($237 \, \mathrm{W \, m^{-1} \, K^{-1}}$) and approaches that of copper ($401 \, \mathrm{W \, m^{-1} \, K^{-1}}$), which are all thermally conductive metals. The thermal transport experiments[28,29] further confirm the efficient heat transfer along the in-plane direction of GFFs (see Supplementary Fig. 3).

In view of the relatively low density of GFFs, roughly $0.22 \, \mathrm{g \, cm^{-3}}$ for 130GFF-3,000 and $0.23 \, \mathrm{g \, cm^{-3}}$ for 200GFF-3,000, we acquired specific electrical conductivity ($\sigma/\rho$) and specific thermal conductivity ($\kappa/\rho$) to make a fair comparison with other materials (Fig. 4b and Supplementary Table 2). Comparing with previous 2D assemblies of nanocarbons, namely carbon nanotube or graphene films/papers[25,30–33], 130GFF-3,000 exhibits far better specific electrical and thermal conductivities. The specific electrical conductivity of GFFs is nearly three times that of commercially available carbon fibre papers, while the specific thermal conductivity is 30 times higher (data were obtained from the website of Toray Industries Inc.). Even when compared with the graphene fibres annealed at 2,850 °C (ref. 7), our GFFs show a two times higher specific thermal conductivity, as well as a comparative specific electrical conductivity. In addition, the electrical and thermal conductivities both increase systematically with the density of 130GFFs, which is in accordance with the case of previously reported porous carbon materials[31,34,35] (Supplementary Fig. 4). These results are indicative of well-balanced conductivity and lightweight in the GFFs, which provide them with great potential as highly electrically and thermally conductive scaffolds.

The outstanding electrical and thermal conductivities of 130GFF-3,000, with no degradation comparing with the individual graphene fibres, are basically attributed to two reasons. Besides the defect-free and crystalline structure of graphene after high-temperature annealing, the junctions within

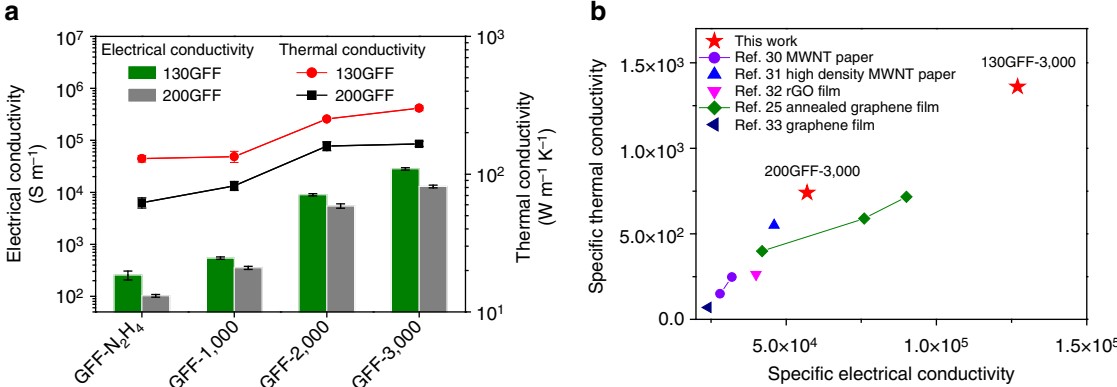

**Figure 4 | Electrical and thermal conductivities of GFFs.** (**a**) In-plane electrical and thermal conductivities of 130GFFs and 200GFFs after $N_2H_4$ reduction and thermal annealing at various temperatures. Error bars represent the s.d. of the conductivities of different GFFs. (**b**) Comparison of specific electrical conductivity ($\sigma/\rho$) and specific thermal conductivity ($\kappa/\rho$) of GFFs with selected 2D assemblies of carbon nanotube (CNT) or graphene. The units for $\sigma$, $\kappa$ and $\rho$ are $S\,m^{-1}$, $W\,m^{-1}\,K^{-1}$ and $g\,cm^{-3}$, respectively.

the interfused fibre network play an important role in ensuring efficient conduction between the highly conductive fibres (Supplementary Fig. 5).

**Microstructures of GOFFs and GFFs.** Unlike the laminated graphene films, GFFs exhibit a hierarchical microstructure of randomly crosslinked fibre network (Fig. 5a–c). Thus, even the GOFF built by hydrophilic GO fibres showed water repellence with a 127.9° contact angle (Supplementary Fig. 6a), higher than that of GO films[36–38], which was normally below 80°. Interestingly, a time-dependent variation of contact angle was observed for both GOFFs and GFFs (Supplementary Fig. 6b).

Thermal annealing caused an obvious decrease in fibre diameter (Supplementary Table 3), whereas the interfused network structure was well-preserved (Fig. 5d,e). In contrast to the large number of fused junctions in 130GFFs, there are much less junctions in 200GFFs (Fig. 5c,f), probably because of the increased difficulty in swelling of thicker fibres. Since the fused junctions connect conductive paths for electrons or phonons via eliminating the contact resistance, it is reasonable to have better conductivities in 130GFFs. In addition, most of the graphene fibres are randomly oriented in the in-plane direction of GFFs, and loosely packed along the out-of-plane direction in both 130GFFs and 200GFFs (Supplementary Fig. 7a,b). The anisotropic nature of both assembled structures and individual graphene fibres results in anisotropic transport properties that the through-plane electrical conductivity of GFFs ($\sim$138.9 $S\,m^{-1}$ for 130GFF-3,000 and $\sim$124.3 $S\,m^{-1}$ for 200GFF-3,000, respectively) is two orders of magnitude lower than the conductivity along the in-plane direction (Supplementary Fig. 7c).

Generally, there are two typical junctions within the fabrics: X-type junctions linking four directions (Fig. 5g) and Y-type junctions linking three directions (Fig. 5h). Others are combinations of these two in different forms (Supplementary Fig. 8), all of which are beneficial to fabric performance. The formation of junctions via fusing is always accompanied by rearranging of graphene sheets within the contact area, evidenced by the indistinct boundary between two fused fibres. Such fully integrated junctions are sufficiently strong to assure efficient load transfer from one fibre to the entire fabric. Even at the broken end after fracture of GFF, the junctions still kept intact (Fig. 5i). The readily accessible interfused structure is actually the most remarkable advantage of GFFs over commercially available carbon fibre papers, where the joints between carbon fibres are known as defects degrading the overall performance.

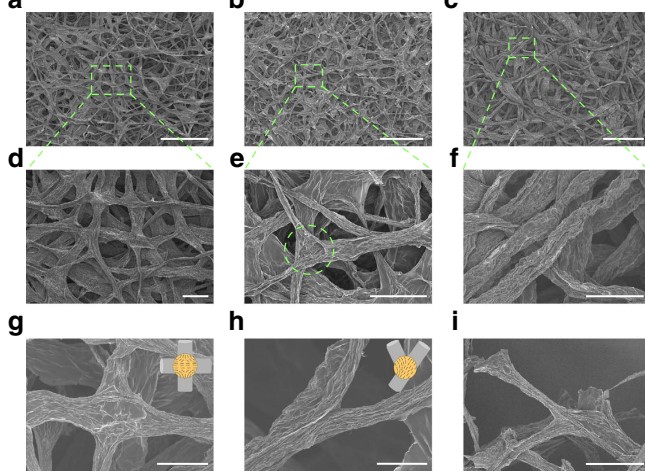

**Figure 5 | Microstructure of GOFFs and GFFs.** SEM images of (**a**) 130GOFF, (**b**) 130GFF-3,000 and (**c**) 200GFF-3,000. (**d–f**) Magnified images of **a–c**. SEM images highlighting the (**g**) X-type junction, (**h**) Y-type junction and (**i**) a well-preserved junction at the broken end of 130GFF-3,000. Insets in **g,h** depict the arrangement of graphene sheets within X-type junction and Y-type junction. Scale bars, 500 μm (**a–c**), 100 μm (**d,e,f,i**) and 50 μm (**g,h**).

**Structural stability and fracture behaviour of GFFs.** The firmly bonded structure ensures adequate mechanical strength for bearing diverse forms of deformation. We first investigated the bending behaviour of GFFs by monitoring the variation of electrical resistance. The resistance of a 130GFF-3,000 was nearly unchanged (0.3% variation) up to a bending radius of 1.5 mm (Fig. 6a), suggesting a high tolerance on bending deformation. After 1,000 bending-releasing cycles for a radius of 2 mm, the resistance kept stable (Fig. 6b) with obscure variation below 1.3%. Even under more violent deformation of folding, which is fatal to most fragile materials, the resistance barely changed after 10 repeated folding-releasing operations (Fig. 6c). The full recovery after folding deformation left no apparent crease on the surface of GFF (inset in Fig. 6c). Combining outstanding conductivity and flexibility simultaneously, the GFFs show promise as flexible conductors.

Uniaxial tensile measurements revealed that 130GFFs are much stronger than 200GFFs (Fig. 6d), mainly because of the more efficient load transfer achieved in the well-connected fibre

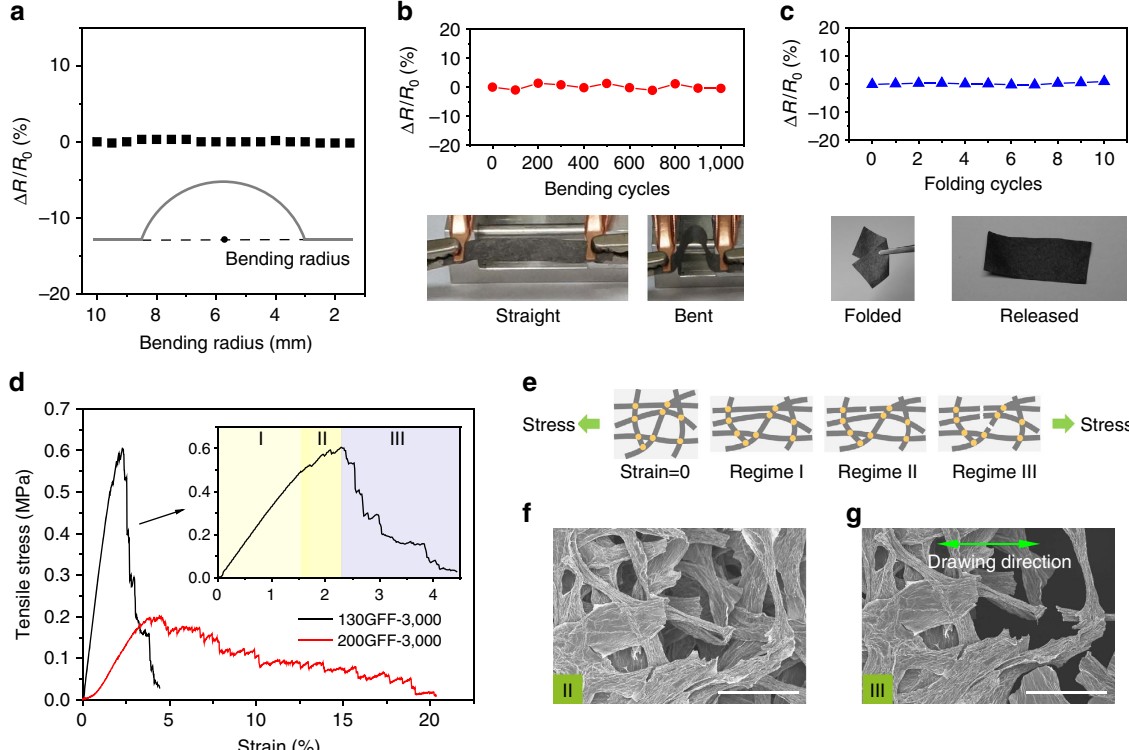

**Figure 6 | Bending and stretching behaviour of GFFs.** Electrical-resistance variation of a GFF (**a**) at bending radius up to 1.5 mm, (**b**) under cyclic bending for 1,000 times and (**c**) performing 10 folding-releasing cycles. $R_O$ is the initial resistance of the GFF and $\Delta R$ is the resistance change in different states. Inset in **a** shows the definition of bending radius. Photos in **b** show photographs of a GFF in straight and bent states, respectively. Photos in **c** show a GFF being folded by a pair of tweezers and released. (**d**) Typical stress–strain curves of 130GFF-3,000 and 200GFF-3,000. Inset emphasizes details of the stress–strain curve of 130GFF-3,000. (**e**) Diagram of the fracture process under tensile stress. SEM images showing crack propagation through the thickness of the GFF in (**f**) regime II and (**g**) regime III. Scale bars, (**f**) and (**g**) 100 μm.

networks. The tensile strength and modulus of 130GFF-3,000 are $0.6 \pm 0.1$ and $29.9 \pm 6.4$ MPa, whereas the values for 200GFF-3,000 are only $0.2 \pm 0.02$ and $5.7 \pm 1.3$ MPa, respectively. The two types of GFFs expressed obvious difference in their fracture behaviour during tensile tests. 200GFFs with a poor interfibre interaction showed evident initial strengthening, large elongation at break and a loose morphology near the broken end (Supplementary Fig. 9), resulting from the slippage between the unfused fibres.

While the graphene fibres within 130GFFs are firmly bonded to guarantee an effective load transfer through the whole stretching process, three regimes of deformation are observed in the stress–strain curve of 130GFF-3,000 (Fig. 6e). In regime I, GFF was stretched along the drawing direction and the stress increased linearly with tensile strain, representing an elastic deformation. The stress–strain curve in regime II is on the fluctuant rise. Similar tensile behaviour in GO papers was explained by a slide-and-lock mechanism where the graphene sheets slide and then click into place when progressively stressed[23,39]. In our circumstance, however, graphene fibres are joined by fused junctions; therefore, the former mechanism is not applicable. The investigation upon crack propagation shows that a crack first generated in a small damage zone (Fig. 6f), and then ran through the entire cross-section under continuous stretching (Fig. 6g). It is a straightforward evidence of a stepwise-breaking mechanism: the constant drawing makes some of the fibres over stretched prior to others, creating instant stress release at the time when fibre breaks. Then, the increasing load is immediately carried by newly stretched fibres to maintain an upward trend of the stress. At last, in regime III, the growing breakage on graphene fibres leads to

fracture of GFF after reaching a maximum load. According to the same stepwise-breaking mechanism, the fracture of GFF is in a gradual manner, rather than a sudden stress drop, which is normal in compact graphene assemblies[23,32].

Generally speaking, the three main factors that determine the performance of GFFs are attributed to the properties of an individual staple fibre, fibre length and the interaction between fibres. Here the graphene staple fibres after 3,000 °C annealing exhibit mechanical strength of ∼21.1 MPa (Supplementary Fig. 10) and electrical conductivity of ∼4.5 × 10⁴ S m⁻¹. The GFFs hold great promise for higher performances since there is large room for substantial improvement on staple fibres[8,40]. Second, the length of staple fibres (2–5 mm) was optimized in our design. When the fibres were longer than 5 mm, it was hard to form a uniform piece of GFF because of the entanglement between fibres. On the other hand, GFFs with shorter fibres (less than 2 mm) exhibited poorer conductivities owing to the reduced conduction within a single fibre (Supplementary Fig. 11 and Supplementary Table 4). At last, we found that graphene fibres without wet-fusing assembly only formed a loose pile of staple fibres rather than an integrated fabric even after high-temperature annealing (Supplementary Fig. 12). Therefore, the wet-fusing assembly is critical to the interfusion between graphene fibres, hence to the fabrication of GFFs.

**Application of GFFs for free-standing electrothermal heaters.** Given carbon-based materials are attractive for energy-efficient electrothermal heaters because of their extraordinary Joule heating performance[41–44], we investigated the electrothermal

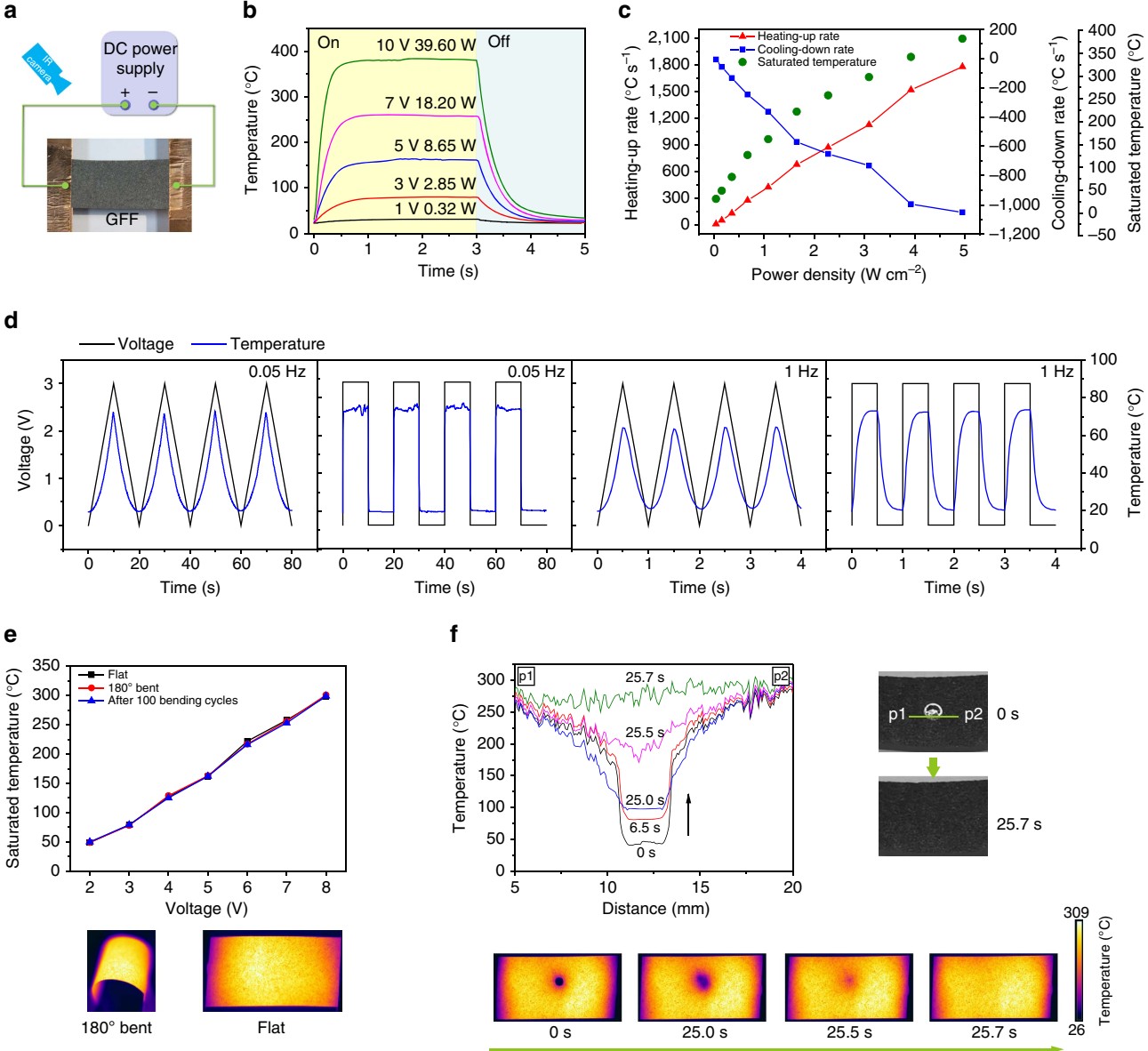

**Figure 7 | Electrothermal performance of GFFs. (a)** Diagram of experimental set-up for GFF electrothermal heaters. **(b)** Temperature profiles of a strip of GFF ($4 \times 2\,cm^2$) at different applied voltages. **(c)** Peak values of heating-up and cooling-down rates and the corresponding saturated temperatures as a function of input electrical power density. **(d)** Frequency-dependent responses of a thinner GFF strip ($20 \times 1.5\,mm^2$) at 0.05 and 1 Hz, with applied triangular wave and pulsed squared wave from 0 to 3 V. **(e)** Top: saturated temperature of a GFF heater at various voltages in flat state, 180° curved state and after bending for 100 times. Bottom: infrared pictures of the GFF heater in flat and 180° bent state. **(f)** Top left: temperature evolution across the central line of a water droplet with respect to time. Top right: photos indicating the concerned region from p1 to p2, and showing the evaporation of the water droplet. Bottom: infrared pictures following the droplet evaporation process.

behaviour of 130GFF-3,000 under ambient condition (Fig. 7a). The free-standing fabric heater was used in large area ($4 \times 2\,cm^2$) and worked at low voltages. The mechanism for heat exchange is discussed in Supplementary Fig. 13 and Supplementary Note 1. As illustrated in Fig. 7b, all the heating processes accomplished within 1 s until the corresponding equilibrium temperatures were reached and the cooling processes cost less than 2 s. While the saturated temperature rises with input power and voltage, the achieved temperatures and heating response for given voltages substantially exceed those of commercial carbon fibre papers, graphite papers (Supplementary Fig. 14), former film Joule heaters, as well as commercial heating elements (Supplementary Table 5). Figure 7c shows that a wide temperature range (30–380 °C) was achieved at low-level working voltages below

10 V. The maximum heating-up rate is linearly proportional to the input power (Fig. 7c and Supplementary Fig. 15), with a noticeable value as high as 1,776.8 °C s$^{-1}$ at 4.95 W cm$^{-2}$ (10 V) and a peak cooling-down rate approaching 1,100 °C s$^{-1}$. When pulsed square wave or triangular wave was employed, the temperature of GFF responded similarly to the input voltage signals at frequencies from 0.05 to 1 Hz (Fig. 7d), which is a further proof of the fast electrothermal response. The fast response of GFFs is attributed to their ultrahigh specific electrical conductivity, since the low electrical resistance could generate a large amount of heat ($Q = U^2/R \cdot t$), while the relatively low mass of GFFs is easier to be heated.

Another advantage of the mechanically robust GFF heaters is their stability when work in bending states. The saturated

temperature kept steady in both flat and 180° curved GFFs. After cyclic bending for 100 times, the difference between temperature plots is still not evident (Fig. 7e), in accordance with the above-mentioned structural stability of GFFs under bending deformation. The infrared thermal images at an applied voltage of 7 V (Fig. 7e) show that the temperature distribution on GFFs is uniform, either in flat or bending state.

The high-performance and large-area GFF heater is capable of evaporating massive water in several seconds (Supplementary Movie 4). Upon a simplified prototype, we followed the heating process on a water droplet. When a direct voltage of 8 V was applied, the elimination of a water droplet (~3 mm in diameter) was accomplished in less than 26 s. The simultaneously recorded thermal images and temperature evolution cross the central line of the droplet (Fig. 7f) tell the story of the evaporation process. Since the steady-state temperature of the bare sample at 8 V is ~300 °C, it took ~25 s to heat the droplet to its boiling point (100 °C). In the next 0.7 s, temperature in the droplet area grew rapidly from 100 to 300 °C, accompanying with instant evaporation of water. Right after fully removal of the droplet, the uniformity of temperature distribution on the GFF recovered immediately.

Collectively, the efficient electric heating, ultrafast electrothermal response, uniform temperature distribution and good flexibility taken together make the GFFs excellent for large-area flexible heaters. In particular, the heating phenomenon on GFFs is evident even at low voltages (~50 °C at 2 V, ~80 °C at 3 V and ~100 °C at 3.7 V), which, in combination with breathability and light weight, is highly attractive for wearable heating elements.

**Application of GFFs for oil-adsorbing felts.** Furthermore, GFFs with hydrophobicity, porous structure and good mechanical property are applicable for practical oil-adsorbing, which showed strong adsorption capability up to 80 times their own weight for a collection of organic solvents, and especially viscous oils (Fig. 8a,d). They exhibited several times higher adsorption capacity than many previously reported 2D adsorbents, for example, wool-based nonwovens (8–14 times weight gain)[45], nanowire membranes (6–20 times weight gain)[46] and commercial polypropylene (PP) oil absorption felts (7–11 times weight gain). More importantly, GFFs are robust enough to withstand violent agitation in liquids (25 cm s$^{-1}$, Fig. 8b). On the contrary, three-dimensional aerogels are easily broken into pieces during a mild agitation, despite their higher adsorption capacity[47–50] (Fig. 8c and Supplementary Movie 5). For practical uses in natural conditions, such as oil clean-up in the ocean, the robustness of a large-area adsorbent is extremely important to sustain the shock from strong winds, waves and water current. Therefore, the mechanically strong GFFs are ideal practical adsorbents for pollutant removal.

Adsorption rate is another criterion for evaluating adsorbents, as it is crucial for fast removal of organic pollutants in real applications. The adsorption in GFF, taken heptane for example, occurred rapidly after inserting a piece of GFF into the liquid. Wicking of heptane driven by capillary force was obviously seen at a rate around 55 mm s$^{-1}$ (Fig. 8e), whereas the wicking rate in a commercial PP felt is only ~4 mm s$^{-1}$. The adsorbed heptane could be eliminated by direct combustion to regenerate the GFF (Supplementary Movie 6). After 20 adsorbing-burning cycles, the adsorption capacity of GFF barely changed (Fig. 8f). Meanwhile, the flexibility of GFF was maintained, showing almost no damage on the fused network structure (Fig. 8g). In a word, the open pores within GFFs provide the capability of efficient and fast adsorption, which is an order of magnitude

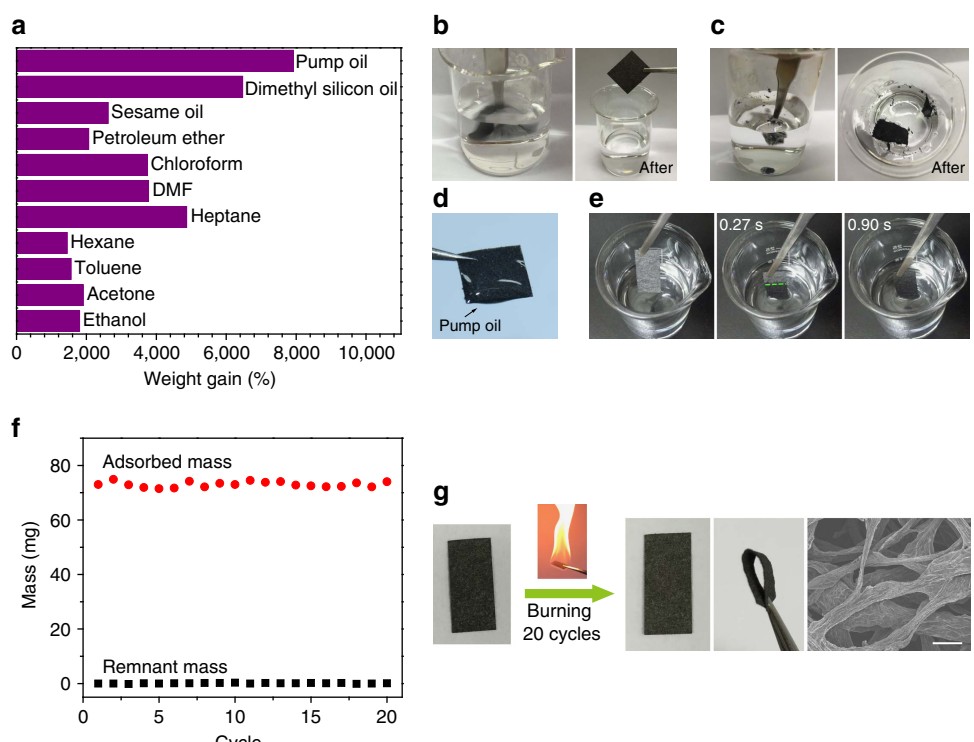

**Figure 8 | Oil uptake behaviour of the GFF.** (**a**) Adsorption capacities of GFF for various organic liquids in term of its weight gain. (**b**) Photos showing violent agitation of a GFF in water, and the GFF remains intact after agitation. (**c**) Mild agitation of a graphene aerogel in water makes the aerogel broken into pieces. (**d**) Photograph of a GFF adsorbing pump oil with relatively high viscosity. (**e**) Fast adsorption of heptane within 1 s. The dashed line indicates the frontline of adsorbed heptane. (**f**) Recyclability of the GFF adsorbing felt. Combustion was applied to regenerate the GFF with adsorbate of heptane. (**g**) The appearance, flexibility and microstructure of GFF are not changed after 20 adsorbing-burning cycles. Scale bar, 50 μm.

higher in both capacity and rate than those of commercial PP felts; at the same time, the interfused framework of GFFs ensures the robustness for easy manipulation, durability and stable recyclability.

## Discussion

We proposed an assembling methodology to make GFFs from as-prepared solid individual graphene fibres. Our method has obvious advantages: first, large-scale productivity since the 'building blocks' of individual graphene fibres with designed diameters and lengths can be massively produced in advance; second, easy controllability in terms of shape, area, thickness and microstructure of individual fibres including compact, core sheath, porous and hybrid ones; and last, solving the key problem of fabric shrinkage during drying. We intentionally focused on the fused junctions of graphene fibres, and found out the significant effect of fused junctions on the mechanical, electrical and thermal properties of GFFs. The fused junctions and fibre network are highly stable during high-temperature annealing, opening the avenue to high-performance GFFs from the defective GO.

The successful fabrication of non-woven GFFs offers an efficient solution to extend the application of graphene fibres from 1D into large-area 2D field. The concept of non-woven GFFs is new to graphene-based assemblies, which is macroscopically paper-like and shows substructure of randomly crosslinked graphene fibres at the microscopic scale. The interspaces among fibres make the GFFs lightweight and provide penetration paths for light, gases, liquids, and so on. Their flexibility is also derived from the interfibre spacing that has the capacity to tolerate fibre deformation. On the other hand, the well-connected fibres effectively support the framework and link transmission routes for electrons and phonons, thus leading to record-breaking-specific electrical and thermal conductivities for 2D assemblies of nanocarbons. Benefiting from the interfused network and an all-graphene structure, our binder-free GFFs outperform commercially available carbon fibre papers on both conductivity and flexibility. In view of the broad application of carbon fibre papers in the green energy fields, such as gas diffusion layers for fuel cells[51,52] and electrodes for energy storage[53–55] and water splitting[56] devices, the GFFs should become an alternative choice showing a great appeal. Our preliminary attempt to build supercapacitors using 130GFF-$N_2H_4$ as electrodes revealed remarkable gravimetric (206 F g$^{-1}$ at a current density of 1 A g$^{-1}$) and areal capacitance (220 mF cm$^{-2}$ at 1 mA cm$^{-2}$; see Supplementary Fig. 16). These results demonstrated excellent electrochemical performance of GFFs, overwhelming commercially available carbon fibre papers[53]. Moreover, the structural units of GFFs are not limited to ordinary compact graphene fibres; a wide variety of derivatives, such as hybrid fibres[57–59], nacre-mimic fibres[60–64], porous fibres[65] and so on, are also available as extensions, allowing the GFFs a versatile substitution for carbon fibre papers.

In conclusion, we demonstrated a facile wet-fusing assembly strategy for production of non-woven GFFs with strong interfibre interaction. The constituent staple fibres are randomly oriented and interfused within the fabrics. The high-quality graphene after thermal annealing and interfused network structure endows GFFs with outstanding mechanical robustness, flexibility, as well as excellent electrical and thermal conductivities. The fabrication process is simple and is able to be a general strategy for GFFs with designed fibre structures and compositions. A wide range of applications can be envisioned for such multifunctional fabrics: electrothermal heaters, oil adsorbents, separator membranes, conductive scaffolds for composites and electrodes, and catalyst supports, to name a few.

## Methods

**Wet-spinning of GO staple fibres.** Aqueous GO solution made by a modified Hummers' method was purchased from C6G6 Co. Ltd. in a concentration of 14.5 mg ml$^{-1}$. The GO sheets are mostly monolayer, with lateral size in the range of 40–50 μm. The GO solution in water was then subjected to a solvent exchange process using N, N-dimethylformamide (DMF). The obtained GO/DMF solution (∼5 mg ml$^{-1}$) was used as the spinning dope for GO staple fibres and injected into a rotating coagulation bath of ethyl acetate. The injection and rotation speeds were set as 40 μl min$^{-1}$ and 40–50 r.p.m., respectively, for a spinneret with diameter of 130 μm. As a reference, a 200 μm spinneret was also employed, correspondingly with an injection speed of 50 μl min$^{-1}$. After 60 min immersion in the coagulation bath, GO stable fibres were collected by vacuum filtration, dried at room temperature for 12 h and then 60 °C under vacuum for 3 h, in order to completely eliminate the solvents in the fibres.

**Preparation of non-woven GOFFs.** The dried GO fibre stack was first put into a mixture of water and ethanol (volume ratio of 3:1) to realize re-dispersion of GO fibres. Then, the re-dispersed GO staple fibres were collected using a plastic mesh. After drying at 80 °C for 10 h, a piece of non-woven GOFF was obtained.

**Conversion of GOFFs into GFFs through chemical reduction/thermal annealing.** Hydrazine reduction was performed by putting the GOFFs into a sealed glass vessel filled with hydrazine vapour and kept at 95 °C for 12 h. Thermal annealing was carried out at 1,000, 2,000 and 3,000 °C with argon protection for 1 h.

**Characterization.** Optical and polarized-light optical micrographs were captured using a Carl-Zeiss AxioCam MRc5 microscopy. X-ray diffraction measurements were taken on a Philips X'Pert PRO diffractometer using Cu Kα1 radiation (40 kV, 40 mA) with an X-ray wavelength ($\lambda$) of 1.5418 Å. Raman spectra were acquired using a Renishaw inVia-Reflex Raman microscopy at an excitation wavelength of 532 nm. XPS was performed using a PHI 5000C ESCA system operated at 14.0 kV. All binding energies were referenced to the C1s neutral carbon peak at 284.8 eV. Scanning electron microscopy (SEM) images were taken on a Hitachi S4800 field-emission SEM system. Contact angles of water on the GOFFs and GFFs were determined using a Dataphysics OCA20 optical instrument at ambient temperature. Gas permeability was measured using a Labthink TQD-G1 Air Permeability Tester under 25 Pa pressure. Electrical conductivity of GFFs was calculated from the slope of I–V curves with a scanning range from −1 to 1 V. Thermal conductivity was measured utilizing a well-established self-heating method at room temperature[7,25,30,32]. At least three measurements were carried out for the average value of electrical and thermal conductivities. Density of the GFFs was determined by the ratio of mass divided by volume. The electrical-resistance variation was investigated by acquiring the I–V curves while bending. Tensile measurements were performed on a Microcomputer Control Electronic Universal Testing Machine made by REGER (RGWT-4000-20), and equipped with a 5 N load cell. The gauge length was 10 mm and the loading rate was set as 1 mm min$^{-1}$. Electrothermal behaviour of GFFs was studied mainly on a rectangular sample (4 × 2 cm$^2$) powered by a d.c. power supply (Gratten APS3005D), while temperature of the sample was monitored using an infrared camera (FLIR T630sc). Frequency-dependent responses of GFF were investigated using a thinner strip (20 × 1.5 mm$^2$) to simplify the experimental set-up. The weight gain of organic liquids was determined by measuring the weight before and after adsorption.

**Data availability.** The data that support the findings of this study are available from the corresponding author upon request.

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

## Acknowledgements

This work was supported by the National Natural Science Foundation of China (Nos 21325417 and 51533008), MOST National Key Research and Development Program (No. 2016YFA0200200) and the State Key Laboratory for Modification of Chemical Fibres and Polymer Materials, Donghua University (No. LK1403).

## Author contributions

Z.L. and C.G. conceived and designed the research, analysed the experimental data and wrote the paper. Z.L. conducted the experiments. Z.X., Y.L. and R.W. took part in discussion on the results and modified the manuscript. C.G. supervised and directed the project.

## Additional information

**Competing financial interests:** The authors declare no competing financial interests.

