## [Peer Review File · Nature Communications]

Reviewers' comments:

Reviewer #1 (Remarks to the Author):

In this paper, wet-spun graphene fibers are vacuum filtrated into a mat structure. Wet-fusing is employed to create junctions between graphene fibers to enhance the structure cohesion and facilitate electrical and thermal transport. The electrothermal heating and oil-adsorbing performance are investigated. The wet-fusing method has been reported to fabricate graphene fiber network (ref 17), which is not new. Besides, the reported mechanical and physical properties are order of magnitude lower than that of commercialized carbon fiber clothes. The fiber mat does not show any unique properties or advantages when being applied as electrical heater and oil adsorbent. No potential applications can be envisioned based on such mat structures. I do not recommend this publication on "Nature communication".

1, the graphene fiber mat in this paper are assembled from short fibers, which are much weaker than the carbon fiber clothes knitted from long fibers with respect to mechanical and physical properties. Inter-fibre interaction is not a dominant issue in case of long fiber knits, actually, heat transport and mechanical reinforcement are more inclined to long range continuous structures, rather than interconnected short fibers. Taking short carbon fibers as a reference, the graphene fiber mat may have potential being used as fillers in composite for thermal and electrical conductivities enhancement, but the expected performance still much inferior when comparing with long fiber knits.

2, Intrinsically, carbon materials with a low electrical resistance are always considered as good electrical heater. Authors are recommended to compare the heating efficiency or response time between GFF and conventional graphite film, carbon fiber clothes and alloy, ceramic heating element, rather than simply claiming the "efficient electrical heating, ultrafast eletrothermal response" without any comparison.

Other concern is when authors characterized the heating or cooling performance, the heat transport mechanism should be considered. Within a wide temperature range (30-380 {degree sign}C) the surface condition and internal porous structure of GFF and the involved heat transfer mechanisms (natural or force convection, irradiation) are all necessary to be considered. Reference 39 is a good example.

3, regarding the oil adsorbing performance, the reported GFF displays a much lower capacity when comparing with form/aerogel dominated oil sorbent, although authors emphasize the mechanical robustness. A higher porosity is favorable for oil adsorbing, however, a trade-off is the sacrifice of mechanical robustness as the weak cohesion. It is believed more porosity can be created in the GFF; correspondingly, electrical and thermal transport, and mechanical robustness will be degraded. Thus, GFF does not demonstrate any advantages over traditional adsorbents.

Reviewer #2 (Remarks to the Author):

The authors developed a wet-fusing technique to create non-woven graphene fibre fabrics (GFFs) composed of interfused graphene fibres with strong inter-fibre interaction. The porous GFFs exhibit superior electrical conductivity (2.8×10^4 S/m) and thermal conductivity (301.5 W/mK) with the very low density (0.22 g/cm³). The GFFs show wide and potential applications such as highly efficient, flexible electro-thermal heater and recyclable sorbent for oils and organic solvents. The concept of wet-fusion for merging the interface between graphene fibres is intuitive and the result is promising, to be able to inspire others to build GFF-based applications, like functional textiles. This work may attract much interest and attention of scientific community; therefore I recommend publication of this article in Nature Commutations after major revision, as shown below:

1. The GFFs with wet-fusing treatment exhibit excellent electrical, thermal, and mechanical properties. The authors should interpret if these properties are anisotropic or isotropic and give the values determined along in- and out-of-plane directions, respectively.

2. The high temperature annealing is a conversional way to eliminate the poor interface between

carbon materials due to the solid state interdiffusion. I suppose the difference in thermal/electrical properties of the samples with and w/o wet-fusing treatment might decrease with the increase of annealing temperature, especially over 2000 °C. The authors should examine and compare the results as described above.

3. The major advantage of using wet-fusing treatment is to improve the mechanical properties of GFFs, but in this article the authors paid less attention to it. The authors should do more studies on the enhancement of mechanical properties of wet-fused GFFs.

4. Considering their low density, the thermal conductivity of annealed samples is quite high (> 300 W/mK), but the property was determined by using a homemade apparatus. The authors can take the IR images to show the comparison of the temperature increase as a function of the heating time between the samples and other materials, such as aluminum (≈ 230 W/mK). Referring to figure 4a in 10.1016/j.compositesa.2015.02.011 or figure 6b in 10.1016/j.compositesa.2016.05.010

Reviewer #3 (Remarks to the Author):

1) Figure 2: The authors show that the thickness of the fabric could be as thick as 3 mm. In this case, what is the morphology in the thickness direction? Is the material isotropic? It is suggested that the SEM image of the cross-section be provided. The authors should also clarify along which direction, in-plane or through-the-thickness, the properties reported were measured and whether these properties differ between the two directions.

2) Figure 4: the electrical and thermal conductivities of GOFF should also be included for comparison in Figure 4a. What do the two color shaded areas in Figure 4b represent?

3) What is the length of individual graphene fibers produced from wet spinning? Does the length of the graphene fibers affect the properties of GFFs? What are the limiting factors for the length of graphene fibers?

4) Graphene fibers made from spinnerets with different diameters are used to fabricate two types of GFFs (i.e. 130GFFs and 200GFFs). It is shown that 130GFFs had better properties than 200GFFs because 200 GFFs did not form as many junctions as 130 GFFs during the assembly process (Figure 5c). Why does larger diameter inhibit the formation of junctions between graphene fibers? Moreover, is the diameter of the graphene fiber an optimized parameter? What if the diameter of the graphene fiber is even smaller than those in 130GFFs?

5) Figure 5: Why are the contact angles of GOFF and GFF important? From the measurement, the GOFF is rather hydrophobic even though GO itself is hydrophilic. Explain why. Besides, the authors also measured the time-dependent contact angles of both GOFF and GFF, both of which decreased with time. What are the implications of this observation? What material characteristics do the authors expect to reveal with a series of contact angle measurements?

6) Figure 6f and g: From which stage of loading are these SEM images taken? The authors need to correspond each image to the stress-strain curve shown in Figure 6d.

7) Page 14: For the application of electrothermal heater, a comprehensive benchmark should be provided between GFFs and others reported in the literature in terms of driving voltage, heating efficiency, response time etc. to show the advantage of using the current material.

Response to Reviewers

For Reviewer #1:

General comment: In this paper, wet-spun graphene fibers are vacuum filtrated into a mat structure. Wet-fusing is employed to create junctions between graphene fibers to enhance the structure cohesion and facilitate electrical and thermal transport. The electrothermal heating and oil-adsorbing performance are investigated. The wet-fusing method has been reported to fabricate graphene fiber network (ref 17), which is not new. Besides, the reported mechanical and physical properties are order of magnitude lower than that of commercialized carbon fiber clothes. The fiber mat does not show any unique properties or advantages when being applied as electrical heater and oil adsorbent. No potential applications can be envisioned based on such mat structures.

Response: Thanks for the comment. We believe that our work is of novelty and the graphene fibre mat shows many potential applications.

The novelty of our work is mainly attributed to four aspects:

- 1) It is the first time to utilize an assembly methodology to make graphene fibre fabrics (GFFs) from as-prepared solid individual graphene fibres. Compared with the direct wet-spinning method to make a mat (ref. 17), our method has obvious advantages: (i) large-scale productivity since the “building blocks” of individual graphene fibres with designed diameters and lengths can be massively produced in advance; (ii) easy controllability in terms of shape, area, thickness, and microstructure of individual fibres including compact, core-sheath, porous, and hybrid ones; (iii) solving the key problem of fabric shrinkage during drying (the wet-spun graphene oxide (GO) fibres would severely shrink at drying, which cannot be overcome by the previously reported method including ref. 17).
- 2) It is the first time to intentionally focus on fused junctions between graphene fibres. Ref. 17 focused on the programmable writing method to make graphene fibre network, accompanied with the observation of wet-welded GO fibres, and it did not pay much attention on investigating the formation mechanism of junctions, integral properties of the network, and practical uses of the network. Therefore, it is our work that finds out the significant effect of fused junctions on the mechanical, electrical and thermal properties of GFFs. We studied in detail the formation

process of junctions, integral properties of the fused GFFs, and potential applications of the fabrics in electrothermal heaters, oil adsorbents, and electrodes for supercapacitors.

Such an intentional research on a known phenomenon is very important to discover new theories, attributes, and potentials. For examples, Newton discovered the law of gravitation from the well-known falling phenomenon, people invented sonar from the inspiration of echolocation of a bat, scientists understood the mechanism of super-hydrophobicity of lotus leaf and then utilized it to create new materials from the well-known phenomenon of “live in the silt but not imbrued”, etc.

- 3) It is the first time to discover the stability of fused junctions and fibre network during high-temperature annealing. High-temperature annealing may cause the expansion and shrinkage of a material, which would destroy the fused junctions and then the fibre network. Nevertheless, we found that the graphene fibre junctions and networks are highly stable in such harsh conditions, opening the avenue to high performance graphene fibre fabrics from the defective GO.
- 4) It is the first time to reveal the excellent properties of fused GFFs especially the prominent electrical and thermal conductivities.

Furthermore, we also demonstrated the potential applications of our fabrics as fast-responsive electrical heater, oil-absorbent and electrodes for supercapacitors, showing much higher performance than the commercial carbon fibre clothes/papers. More applications can be envisioned on the basis of the natures of GFFs, such as separator membranes, conductive scaffolds for composites, catalyst supports, electrodes for energy conversion and storage devices, etc.

Of course, new things are not always perfect but promising. We believe that the properties especially the mechanical properties of GFFs can be highly improved and more applications can be widely demonstrated in the future.

Comment 1. The graphene fiber mat in this paper are assembled from short fibers, which are much weaker than the carbon fiber clothes knitted from long fibers with respect to mechanical and physical properties. Inter-fibre interaction is not a dominant issue in case of long fiber knits, actually, heat transport and mechanical reinforcement are more inclined to long range continuous structures,

rather than interconnected short fibers. Taking short carbon fibers as a reference, the graphene fiber mat may have potential being used as fillers in composite for thermal and electrical conductivities enhancement, but the expected performance still much inferior when comparing with long fiber knits.

Response: Thanks for the comment and suggestion. Non-woven fabrics constructed by short fibres are an important part in the category of fabrics/textiles, having the advantages of easy fabrication and so on. Taking carbon fibre fabrics for example, there are carbon fibre clothes made from long carbon fibres and carbon fibre papers made from chopped carbon fibres. In our current work, we focus on the non-woven graphene fibre fabrics assembled from short graphene fibres, which are never reported in graphene-based macroscopic assemblies. The non-woven configuration is convenient to reach an adequate packing density thus facilitate the integral fabric performance, and is technically viable for industrial production of GFFs. The wet-fusing assembly technique is the key to realize the fabrication of non-woven GFFs while endowing the as-prepared GFFs with many attractive features. We propose the concept of wet-fusing assembly for short graphene fibres, it is also applicable to long graphene fibres, outstanding mechanical and physical properties are highly expected in such graphene fibre clothes as the reviewer suggested. In fact, we are doing such work to extend the wet-fusing assembly on continuous graphene fibres.

Besides the use as reinforcing fillers in composites, carbon fibre papers have broad applications in energy fields, such as gas diffusion layers for fuel cells and electrodes for energy storage and water splitting devices. Our GFFs with better conductivities and flexibility should become an alternative choice showing a great appeal in such areas. In the revised manuscript, we demonstrate the much better performance of GFFs over carbon fibre papers in applications as joule heaters and electrodes for supercapacitors.

Moreover, one of the most promising advantages of our GFFs is the controllable structure of graphene staple fibres, which could give designed functionalities to the fabrics. As we described in the manuscript, the staple fibres which construct GFFs are not limited to ordinary compact graphene fibres. For now, we have nacre-mimic, porous and composite graphene fibres, all of which could be used to assemble the GFFs with various functions. GFFs with designed fibre structures could be applied with great potential, in much broader areas as compared with carbon fibre papers. Therefore,

fabrication of the whole new graphene fibre fabrics should lead to a new research direction of multifunctional carbon-based fabrics.

Comment 2. Intrinsically, carbon materials with a low electrical resistance are always considered as good electrical heater. Authors are recommended to compare the heating efficiency or response time between GFF and conventional graphite film, carbon fiber clothes and alloy, ceramic heating element, rather than simply claiming the "efficient electrical heating, ultrafast electrothermal response" without any comparison.

Other concern is when authors characterized the heating or cooling performance, the heat transport mechanism should be considered. Within a wide temperature range (30-380 °C) the surface condition and internal porous structure of GFF and the involved heat transfer mechanisms (natural or forced convection, irradiation) are all necessary to be considered. Reference 39 is a good example.

Response: Thanks for the suggestions. Comparison of the electrothermal performance between GFFs and commercial carbon fibre papers and graphite papers were carried out under the same experimental conditions. Results are shown in Supplementary Figure 12. The comparison with previously reported film heaters and commercial heating elements is also provided in Supplementary Table 5. The joule heating performance of GFFs is at the top level in terms of high efficiency and quick response.

To understand the mechanism of heat exchange between the GFF heater and the surroundings, we established the heat balance equations for the three stages of heating-up, steady state and cooling-down, respectively. The surface area of the GFF sample is utilized for calculations in a simplified model, since the internal pores are in the range of tens of micrometres thus most of the heat exchange takes place on the surface. At last, we obtained the theoretical radiative and convective heat loss at various temperatures through calculation (Supplementary Figure 11).

Comment 3. Regarding the oil adsorbing performance, the reported GFF displays a much lower capacity when comparing with foam/aerogel dominated oil sorbent, although authors emphasize the mechanical robustness. A higher porosity is favorable for oil adsorbing, however, a trade-off is the sacrifice of mechanical robustness as the weak cohesion. It is believed more porosity can be created

in the GFF; correspondingly, electrical and thermal transport, and mechanical robustness will be degraded. Thus, GFF does not demonstrate any advantages over traditional adsorbents.

Response: Thanks for the comment, which just in turn proves the multi-functionality of our GFFs. The microstructure and then properties can be adjusted in the GFFs, which provides more choice to make fabrics with designed structures and specific application purpose.

Like the reviewer said, there is a balance between the oil adsorbing capacity and mechanical robustness, which is reasonable in most materials. It is right that a higher porosity is favourable for oil adsorbing, however, it is not applicable in practice if the mechanical robustness of the adsorbent is too weak. According to this point, we attained the optimal balance between the adsorbing capacity and mechanical robustness in our GFFs. Therefore, the GFFs not only exhibit superior electrical and thermal conductivities and adequate mechanical robustness, but also show strong adsorption capability for organic solvents and oils, which is not seen in traditional adsorbents.

Above all, our GFFs are actually 2D materials, it is fairer to make comparison with 2D adsorbents. As a result, both the adsorption capacity and adsorption rate of our GFFs are more than 10 times higher than previous 2D adsorbents. The purpose of making further comparison with 3D adsorbents is to demonstrate that our GFFs should be a good choice between 3D adsorbents (higher capacity and lower robustness) and 2D adsorbents (lower capacity and higher robustness). In other words, the GFFs are promising in practical uses while being robust and efficient at the same time.

For Reviewer #2:

General comment: The authors developed a wet-fusing technique to create non-woven graphene fibre fabrics (GFFs) composed of interfused graphene fibres with strong inter-fibre interaction. The porous GFFs exhibit superior electrical conductivity (2.8×10^4 S/m) and thermal conductivity (301.5 W/m K) with the very low density (0.22 g/cm^3). The GFFs show wide and potential applications such as highly efficient, flexible electro-thermal heater and recyclable sorbent for oils and organic solvents. The concept of wet-fusion for merging the interface between graphene fibres is intuitive and the result is promising, to be able to inspire others to build GFF-based applications, like functional textiles. This work may attract much interest and attention of scientific community; therefore I recommend publication of this article in Nature Communications after major revision, as

shown below.

Response: Thanks for the comment and appreciation.

Comment 1. The GFFs with wet-fusing treatment exhibit excellent electrical, thermal, and mechanical properties. The authors should interpret if these properties are anisotropic or isotropic and give the values determined along in- and out-of-plane directions, respectively.

Response: Thanks for the comment. The GFFs exhibit anisotropic electrical, thermal and mechanical properties, which are mainly determined by the anisotropic structure of graphene fibre. As shown in Supplementary Fig. 5a and b, most of the graphene fibres are randomly oriented within the plane of GFFs, and loosely packed along the out-of-plane direction. Meanwhile, the structural units of graphene fibres are also anisotropic because of the well-aligned 2D graphene sheets along the axial direction. When such fibres were stacked into an anisotropic network structure, the integral properties of the fabric will be anisotropic too. Of course, the well fused fibre fabrics show much better performance than the poorly fused ones along the out-of-plane direction.

Since the thermal conductivities and mechanical properties along the out-of-plane direction are hard to be measured for our GFFs, due to their porous morphology and very thin thickness (tens of micrometres to several millimetres), we investigated the through-plane electrical conductivities by measuring the resistances along the thickness direction to verify the anisotropy of GFFs. The values are provided in Supplementary Fig. 5c. Significantly, our GFFs show anisotropic electrical conductivity, and the electrical conductivities along through-plane direction are two orders of magnitude lower than those along in-plane direction.

Comment 2. The high temperature annealing is a conversional way to eliminate the poor interface between carbon materials due to the solid state interdiffusion. I suppose the difference in thermal/electrical properties of the samples with and w/o wet-fusing treatment might decreases with the increase of annealing temperature, especially over 2000 °C. The authors should examine and compare the results as described above.

Response: Thanks for the suggestion. The sample without wet-fusing treatment showed the appearance of a loose pile of graphene staple fibres (Supplementary Fig. 10). The fibres are

independent of each another even after 3000 °C annealing, showing no obvious solid state interdiffusion. Since the fibres were not interfused, a GFF structure was not able to be obtained.

In fact, the 200GFFs are typical examples of poorly interfused GFFs with much less junctions. The solid state interdiffusion is still not observed under SEM observation (Fig. 5c and f), and its influence on thermal/electrical properties of the samples is not evident either, as the results shown in Fig. 4a. We suppose that the relatively low packing density of graphene fibres in our GFFs may limit the solid state interdiffusion while annealing, thus reduce its impacts on the integral properties.

Generally speaking, if the solid state interdiffusion occurs, it could also work in the well interfused GFF samples. Therefore the effects caused by diffusion on fused and unfused GFFs would cancel out, and the principal determinant for electrical/thermal properties is still the number of interfused junctions.

Comment 3. The major advantage of using wet-fusing treatment is to improve the mechanical properties of GFFs, but in this article the authors paid less attention to it. The authors should do more studies on the enhancement of mechanical properties of wet-fused GFFs.

Response: Thanks for the suggestion. In the revised manuscript, we investigate the three factors that influence the properties, including the mechanical properties, of GFFs, which are properties of an individual staple fibre, fibre length, and the inter-fibre interaction (Page 14 in the manuscript, Supplementary Fig. 8-10 and Supplementary Table 4). In our study, the length of graphene fibre was optimized in the range of 2-5 mm, and wet-fusing was employed to enhance the inter-fibre interaction. Currently, the graphene staple fibres are not as strong as the continuous fibres reported by our group (ref. 8), which becomes the main reason that limits the mechanical properties of GFFs. The mechanical properties of staple fibres could be improved if appropriate strengthening protocols were utilized during fibre spinning, according to our previous work, thus the GFFs hold great promise for higher performances by using strengthened staple fibres. However, the fusion between graphene fibres without degradation on their performance is still a big challenge. Our present work offers solution to the wet-fusing assembly of graphene fibres and achieves the GFFs successfully. It is the first step of the research on GFFs. We are trying to improve the mechanical property of GFFs in our on-going work.

Comment 4: Considering their low density, the thermal conductivity of annealed samples is quite high (> 300 W/m K), but the property was determined by using a homemade apparatus. The authors can take the IR images to show the comparison of the temperature increase as a function of the heating time between the samples and other materials, such as aluminum (≈ 230 W/m K). Referring to figure 4a in 10.1016/j.compositesa.2015.02.011 or figure 6b in 10.1016/j.compositesa.2016.05.010.

Response: Thanks for the suggestion. According to the reviewer's suggestion, we provide the IR images showing the comparison of the temperature rise as a function of the heating time between 130GFF-3000 and aluminium foil (Supplementary Fig. 3), which intuitively demonstrates that the thermal conductivity of 130GFF-3000 is higher than that of aluminium indeed.

For Reviewer #3:

Comment 1: Figure 2: The authors show that the thickness of the fabric could be as thick as 3 mm. In this case, what is the morphology in the thickness direction? Is the material isotropic? It is suggested that the SEM image of the cross-section be provided. The authors should also clarify along which direction, in-plane or through-the-thickness, the properties reported were measured and whether these properties differ between the two directions.

Response: Thanks for the comment and suggestions. Please also see the response 1 for Review #2. The SEM images of the cross-section of GFFs are provided in Supplementary Fig. 5a and b. Most of the graphene fibres are randomly oriented in the in-plane direction of GFFs, and loosely packed along the out-of-plane direction in both 130GFFs and 200 GFFs, thus the GFFs are anisotropic. The results displayed in Fig. 4a are in-plane conductivities, which is clarified in the revised manuscript. The through-plane electrical conductivity of GFFs are measured and provided in Supplementary Fig. 5, which is two orders of magnitude lower than the values along in-plane direction, showing an obvious anisotropic feature on the transport properties.

Comment 2: Figure 4: the electrical and thermal conductivities of GOFF should also be included for comparison in Figure 4a. What do the two color shaded areas in Figure 4b represent?

Response: Thanks for the suggestion. The electrical conductivities of GOFFs are extremely low,

nearly 5 orders of magnitude lower than the GFFs-N₂H₄, which are $\sim 2.0 \times 10^{-3} \text{ S m}^{-1}$ for 130GFFs and $\sim 1.5 \times 10^{-3} \text{ S m}^{-1}$ for 200GFFs, we think the values are too low to be included in Figure 4a. Instead, the electrical conductivity of 130GFF is listed in Supplementary Table 4. Also the self-heating method for measuring thermal conductivity in our study is not applicable on GFFs due to their high resistance. The shaded areas in Figure 4b are indication of that the covered data points are acquired in the same reference, in order to avoid misunderstanding, we removed the shaded areas and connected the relevant data points by lines.

Comment 3. What is the length of individual graphene fibers produced from wet spinning? Does the length of the graphene fibers affect the properties of GFFs? What are the limiting factors for the length of graphene fibers?

Response: Thanks for the comment. During the wet-spinning process for GO fibres, the injecting speed for spinning dope, rotating speed of coagulation bath and the position of spinning nozzle are the key parameters determine the length of as-spun fibres. Since the nozzle position is usually fixed, the length of GO fibres could be controlled to some extent through adjusting the speed ratio of injection to rotation. In Supplementary Fig.1, we demonstrate the control over the fibre length during fibre spinning. Normally, the length of individual graphene fiber produced from wet-spinning is ~ 15 mm. After first drying and re-dispersing, the length of fibres may reduce to 2-5 mm. According to our discussion in the manuscript, when the fibres were longer than 5 mm, it was difficult to form a uniform piece of GFF due to the entanglement between fibres. On the other hand, GFFs from shorter fibres (less than 2 mm) exhibited poorer conductivities. Therefore, the length of 2-5 mm is the optimum value for fabrication of GFFs.

Comment 4. Graphene fibers made from spinnerets with different diameters are used to fabricate two types of GFFs (i.e. 130GFFs and 200GFFs). It is shown that 130GFFs had better properties than 200GFFs because 200 GFFs did not form as many junctions as 130 GFFs during the assembly process (Figure 5c). Why does larger diameter inhibit the formation of junctions between graphene fibers? Moreover, is the diameter of the graphene fiber an optimized parameter? What if the diameter of the graphene fiber is even smaller than those in 130GFFs?

Response: Thanks for the comment. We suppose that GO fibres with larger diameter and smaller specific surface area will be more difficult to swell thoroughly due to the increased resistance for water penetration into fibres. Therefore the inter-fibre fusion based on fibre swelling should be inhibited for fibres with larger diameter. The thinner fibres from 130 μm spinneret are preferred for GFFs in our current study according to the higher performance of 130GFFs. Theoretically, utilizing graphene fibres with smaller diameter will benefit the performance of GFFs, owing to the enhanced properties of individual fibres as well as the stronger inter-fibre interaction resulting from the enlarged specific surface area. Considering the efficiency and difficulty of fabrication, however, the current diameter of graphene fibres for 130GFFs is optimized in our present work.

Comment 5. Figure 5: Why are the contact angles of GOFF and GFF important? From the measurement, the GOFF is rather hydrophobic even though GO itself is hydrophilic. Explain why. Besides, the authors also measured the time-dependent contact angles of both GOFF and GFF, both of which decreased with time. What are the implications of this observation? What material characteristics do the authors expect to reveal with a series of contact angle measurements?

Response: Thanks for the questions. The hydrophobicity of GOFFs may originate from the hierarchical microstructure of GOFFs on the surface, even though the GO is hydrophilic. We demonstrate such character of GOFFs in order to emphasize their difference with former compact graphene films, which is essentially determined by the microstructures. The time-dependent contact angles of GOFF and GFF could interpret two things: first, the GOFF and GFF are porous; second, water can penetrate into the fabrics whether they are made by hydrophilic fibres or not. Since the GFFs are promising for energy storage devices, such as supercapacitors and batteries, it is important to know their affinity with aqueous/non-aqueous electrolyte. Therefore the contact angle measurements are necessary for such purposes.

Comment 6. Figure 6f and g: From which stage of loading are these SEM images taken? The authors need to correspond each image to the stress-strain curve shown in Figure 6d.

Response: Thanks for the suggestion. Figure 6f corresponds to the regime II and Figure 6g corresponds to the regime III in the stress-strain curve, respectively. We have added marks in the

corner of each figure.

Comment 7. Page 14: For the application of electrothermal heater, a comprehensive benchmark should be provided between GFFs and others reported in the literature in terms of driving voltage, heating efficiency, response time etc. to show the advantage of using the current material.

Response: Thanks for the suggestion. Comparison of the electrothermal performance between GFFs and commercial carbon fibre papers, graphite papers, previously reported film heaters as well as commercial heating elements is provided in our revised manuscript (Supplementary Fig. 12 and Supplementary Table 5). Both the achieved temperature at various driving voltages and response time were compared, showing great advantage of GFFs as electrothermal heaters.

Reviewers' comments:

Reviewer #1 (Remarks to the Author):

The efforts made by the authors to address the reviewer's concerns are highly appreciated. Specifically, more data were added to compare the electrothermal response between graphene fibers and other materials. However, the results reported are not state-of-the-art as compared with the current fields. The authors agreed that the fused graphene mat is much weaker than long carbon fiber knit, and shows lower oil adsorption capacity as compared with 3D materials despite that it is better than other 2D materials. The authors didn't convey convinced messages on potential applications, and argued that "Such an intentional research on a known phenomenon is very important to discover new theories, attributes, and potentials". It is not clear what the "intentional research" means here. It is true that the scientific breakthrough is based upon many incremental advancements and this can be applied on any scientific research. I don't see how this can justify the publication of this manuscript in Nature Communications. It will be a different story if the advancement can be achieved with new materials, structures or superior properties to the current state-of-the-art.

On the other hand, the key observation of the excellent specific thermal conductivity is questionable. The specific thermal conductivity of 130GFF-3000 ($301.5\text{W/mK}/0.22\text{g/cm}^3 = 1370$), is much higher than that of pyrolytic graphite or pitch-based carbon fiber ($2000\text{w/km}/2.2\text{g/cm}^3 = 909$). Should we achieve over 3000 w/m/k thermal conductivity with a full density of 2.2 g/cm³, which is about 50% higher than HOPG? How could this happen? This is not a single layer graphene.

Reviewer #2 (Remarks to the Author):

In this version, the authors have fully answered the comments raised in my previous review. Therefore, I have no more concern for the publication of this work in Nature Communions. I agree the claim of the reviewer 1. Surely, so far the physical property of GFFs is not comparable to the carbon fiber clothes. However, the low-cost, simplicity, and flexibility of wet-spinning/fusing method enable GFFs as a promising replacement for carbon fiber chopped strand. Considering the short history of graphene research, it can be expected that the performance of GFFs will be further improved.

Reviewer #3 (Remarks to the Author):

While the majority of my comments is well addressed in the revision, the reviewer does have one additional comment following Comment #5. The reviewer suggests the below minor revision before accepting for publication in Nature Communications.

The reviewer is not convinced of the necessity for the time-dependent contact angle measurement. The authors claimed in the response that the aim of measuring the contact angle was to demonstrate the porous structure and water penetration. However, such features were demonstrated more straightforwardly by the SEM images. The authors also mentioned the applications for energy storage devices, which requires the information of affinity between GFF and electrolyte. In this case, what is the significance of time-dependent properties? The authors should further justify the necessity for the time-dependent contact angle measurements with due references in the manuscript.

Point-by-point Response to Referees of

NCOMMS-16-13527A

The revised manuscript entitled “Multifunctional non-woven fabrics of interfused graphene fibres” (NCOMMS-16-13527A) has been further revised according to the comments of reviewers. The point-by-point response is given below.

For Reviewer #1:

General comment: The efforts made by the authors to address the reviewer's concerns are highly appreciated. Specifically, more data were added to compare the electrothermal response between graphene fibers and other materials. However, the results reported are not state-of-the-art as compared with the current fields. The authors agreed that the fused graphene mat is much weaker than long carbon fiber knit, and shows lower oil adsorption capacity as compared with 3D materials despite that it is better than other 2D materials. The authors didn't convey convinced messages on potential applications, and argued that "Such an intentional research on a known phenomenon is very important to discover new theories, attributes, and potentials". It is not clear what the "intentional research" means here. It is true that the scientific breakthrough is based upon many incremental advancements and this can be applied on any scientific research. I don't see how this can justify the publication of this manuscript in Nature Communications. It will be a different story if the advancement can be achieved with new materials, structures or superior properties to the current state-of-the-art.

Response: Thanks for the comment. We have already stated in our last response to the referees that our work is, 1) the first report on macroscopic assembly of non-woven graphene fibre fabrics with outstanding integral performance, 2) the first intensive investigation focusing on the fused junctions between graphene fibres, 3) the first time to discover the stability of fused junctions and fibre network during high temperature annealing, 4) the first time to reveal the excellent properties of fused GFFs especially the prominent electrical and thermal conductivities. The flexibility, specific electrical and thermal conductivities, Joule heating performance and electrochemical

performance of our GFFs are all far better than commercial carbon fibre papers. Especially, the specific electrical and thermal conductivities of our GFFs are much higher than previous 2D assemblies of nanocarbons. The reviewer agrees that the specific thermal conductivity is excellent, as it is pointed out in Comment #2. Meanwhile, the Joule heating performance of GFFs is at the top level in terms of high efficiency and quick response, as compared with previously reported film heaters and commercial heating elements. Therefore, we believe that the novelty and properties of our GFFs are state-of-the-art in the field of high-performance fabrics. Furthermore, a wide range of potential applications have also been demonstrated and envisioned in the manuscript such as electrothermal heaters, practical oil-adsorbents, electrodes for supercapacitors and batteries, etc.

Although the mechanical properties of GFFs are not quite high at present, they could be dramatically improved in the future, as we inferred in the manuscript. Both Reviewer #2 and Reviewer #3 identify with our work while Reviewer #2 provided supportive comments “Surely, so far the physical property of GFFs is not comparable to the carbon fiber clothes. However, the low-cost, simplicity, and flexibility of wet-spinning/fusing method enable GFFs as a promising replacement for carbon fiber chopped strand. Considering the short history of graphene research, it can be expected that the performance of GFFs will be further improved.”

The “intentional research” means we intentionally focused on the fused junctions between graphene fibres and performed detailed study on the formation mechanism and stability during high-temperature annealing of the junctions, as well as their significant effect on the mechanical, electrical and thermal properties of GFFs, which is never reported by others.

At last, as the reviewer said “It will be a different story if the advancement can be achieved with new materials, structures or superior properties to the current state-of-the-art”, the graphene fibre fabric we created here is such a new material with superior flexibility and electrical and thermal conductivities to the current state-of-the-art.

Comment 1: On the other hand, the key observation of the excellent specific thermal conductivity

is questionable. The specific thermal conductivity of 130GFF-3000 ($301.5\text{W/mK}/0.22\text{g/cm}^3 = 1370$), is much higher than that of pyrolytic graphite or pitch-based carbon fiber ($2000\text{w/km}/2.2\text{g/cm}^3 = 909$). Should we achieve over 3000 w/mk thermal conductivity with a full density of 2.2 g/cm^3 , which is about 50% higher than HOPG? How could this happen? This is not a single layer graphene.

Response: Thanks for the comment. As the reviewer points out, “the excellent specific thermal conductivity” is exactly the state-of-the-art feature of our GFFs, which answers the question in the general comment.

On the excellent specific thermal conductivity, more explanations are given below.

1) Besides the measured data, the thermal conductivity of 130GFF-3000 has been verified in our manuscript by comparing the thermal transport evolution with Al foil under infrared camera (the thermal conductivity of 130GFF-3000 is higher than that of Al foil, Supplementary Fig. 3).

2) The thermal conductivity of a material is not only related to its density, but also related to its structure. That’s the reason for different carbon materials exhibiting different specific conductivities, the reason why the specific thermal and electrical conductivities of our GFFs are much higher than those of other 2D nanocarbon assemblies, and also the reason for higher specific conductivities in 130GFFs than 200GFFs, while the two GFFs have similar densities. On the other hand, the specific thermal conductivity for certain material may vary when its density changes. In Ref. 31 (Zhang, L. et al., Nano Lett. 12, 4848-4852 (2012)), the specific thermal conductivity of CNT buckypapers at density of 0.81 g cm^{-3} is 582.7, whereas the specific thermal conductivity decreases to 551.1 when the density increases to 1.39 g cm^{-3} . So the estimation leaving out the effect of structure and simply based on one single parameter of density might not be accurate.

The structure of our GFFs is distinct from that of common porous graphene aerogels, while the latter is actually in the form of expanded graphene laminates. Our GFFs are constructed by plenty of compact graphene fibres which are well connected by fused junctions. It is the fibrous structural units and their interfused configurations that greatly facilitate electrical and thermal conductance along the well-established conducting paths. Although the global packing density of GFFs is low,

the local structure of graphene fibres is still intact, thus the conduction could predominantly and effectively occur within the fibre network. As a result, the specific conductivities of GFFs are extremely high as compared with other 2D nanocarbon assemblies. We suppose that the advantage of GFFs in specific conductivities should be more significant at low densities as the structural difference between GFFs and other materials are diminishing while the density increases to a full density of 2.2 g/cm³.

3) We would like to provide more data on the thermal conductivity of carbon materials for review only (see review only materials).

For Reviewer #2:

General comment: In this version, the authors have fully answered the comments raised in my previous review. Therefore, I have no more concern for the publication of this work in Nature Communions. I agree the claim of the reviewer 1. Surely, so far the physical property of GFFs is not comparable to the carbon fiber clothes. However, the low-cost, simplicity, and flexibility of wet-spinning/fusing method enable GFFs as a promising replacement for carbon fiber chopped strand. Considering the short history of graphene research, it can be expected that the performance of GFFs will be further improved.

Response: Thanks a lot for the appreciation.

For Reviewer #3:

General comment: While the majority of my comments is well addressed in the revision, the reviewer does have one additional comment following Comment #5. The reviewer suggests the below minor revision before accepting for publication in Nature Communications.

The reviewer is not convinced of the necessity for the time-dependent contact angle measurement. The authors claimed in the response that the aim of measuring the contact angle was to demonstrate the porous structure and water penetration. However, such features were

demonstrated more straightforwardly by the SEM images. The authors also mentioned the applications for energy storage devices, which requires the information of affinity between GFF and electrolyte. In this case, what is the significance of time-dependent properties? The authors should further justify the necessity for the time-dependent contact angle measurements with due references in the manuscript.

Response: Thanks for the comment and suggestion. Originally, we observed the phenomenon of time-dependent contact angle for our GFFs, which is in accordance with the results of porous structure. Based on the comment, we carefully re-considered the importance of this observation. Indeed, we agree with the reviewer's comment, and it is not necessary to specifically emphasize this phenomenon in this figure and text. So we moved the contact angle measurement and time-dependent characterization into supplementary information (Supplementary Fig. 5). Instead, the magnified SEM image of 130GOFF was provided in Fig. 5d.

Reviewers' comments:

Reviewer #2:

Reviewer #1 has a significant concern for the extraordinary high thermal conductivity of GFFs. After a simple calculation, I find the thermal diffusivity of either 130GFF-3000 or the carbon material that the authors mentioned in "Review Only Materials" is around $19 \text{ cm}^2/\text{s}$, which is even higher than isotopically enriched diamond ($18.5 \text{ cm}^2/\text{s}$) (10.1103/PhysRevB.42.1104). However, such a high property was measured by a home-made system utilizing so-called well-established self-heating method. Therefore, I recommend that the authors should provide the measurement data for thermal conductivity of GFFs to answer the concern of Reviewer #1. For example, a plot of Signal (V) as a function of Time (ms) for evaluating the thermal diffusivity of the samples can be obtained by laser flash analysis.

Reviewer #3:

Reviewer #1 raised two remarks. I will give my personal opinions on each of them, as follows.

(1) "The results reported are not state-of-the-art as compared with the current fields...."

This statement is quite subjective. However, I assume the reviewer #1 looks into the individual data presented, instead of the overall picture of the paper in a more holistic way.

The nonwoven GFFs developed in this work are conceptually new and indeed possess multifunctional capabilities with exceptional transport properties, permeability, flexibility and light weight, which are demonstrated to be useful as electrodes for supercapacitors and batteries, sensitive electrothermal heaters and durable oil-adsorbing felts. In view of this, the paper has sufficient merits on its originality and overall quality to recommend its publication in the Journal.

(2) "The key observation of the excellent specific thermal conductivity is questionable."

The thermal conductivity of 301.5 W/mK for the GFF with a density of 0.22 g/cm^3 is indeed exceptionally high. Nevertheless, I do not find the result too surprising for the following reasons:

(i) GFF was heat-treated at a very high temperature of 3000 degree C , which greatly helped the reduction of GO. Such high-temperature treatments drastically improve the thermal conductivities of carbon materials, e.g. pitch-based carbon foam (Carbon 2000, 38, 953–973) and graphene papers (Adv. Funct. Mater. 2014, 24, 4542–4548), because of the restoration of the graphitic structure. Such a high degree of graphitization is supported by the extremely high electrical conductivity of 280 S/cm and the Raman spectra.

(ii) The interfused junction can significantly reduce the interface resistance, which also greatly improve the thermal conductivity.

(iii) The fibers were aligned in the plane along which the thermal conductivity was measured. This can also lead to a high thermal conductivity.

However, I also agree with Reviewer #1 in that a higher density can result in a higher thermal conductivity, especially for the porous materials (see Carbon 2000, 38, 953–973; Nano Lett. 2012, 12, 2959–2964). One possible way for the authors to further confirm the validity of the result is to measure the thermal conductivities of GFFs with different packing densities of graphene fibers so that the thermal conductivities indeed increase systematically with the density of GFF. The original density of carbon, 2.2 g/cm^3 , may not be achieved, but at least a trend or correlation can be established which is informative for the validity of the thermal conductivity results.

Point-by-point Response to Referees

For Reviewer #2:

Comment: Reviewer #1 has a significant concern for the extraordinary high thermal conductivity of GFFs. After a simple calculation, I find the thermal diffusivity of either 130GFF-3000 or the carbon material that the authors mentioned in "Review Only Materials" is around $19 \text{ cm}^2/\text{s}$, which is even higher than isotopically enriched diamond ($18.5 \text{ cm}^2/\text{s}$) (10.1103/PhysRevB.42.1104). However, such a high property was measured by a home-made system utilizing so-called well-established self-heating method. Therefore, I recommend that the authors should provide the measurement data for thermal conductivity of GFFs to answer the concern of Reviewer #1. For example, a plot of Signal (V) as a function of Time (ms) for evaluating the thermal diffusivity of the samples can be obtained by laser flash analysis.

Response: Thanks for the comment and suggestion. As the reviewer mentioned, the calculation for thermal diffusivity ($\kappa = \alpha \cdot \rho \cdot c$, where κ , α , ρ and c are thermal conductivity, thermal diffusivity, density and specific heat capacity, respectively) is highly determined by the choice of the value c . In fact, we have measured the c of $3000 \text{ }^\circ\text{C}$ annealed graphene films ($0.76 \text{ J g}^{-1} \text{ K}^{-1}$) by DSC. Then the obtained thermal diffusivity of 130GFF-3000 or the mentioned carbon materials should be around $18.0 \text{ cm}^2 \text{ s}^{-1}$, still lower than that ($18.5 \text{ cm}^2 \text{ s}^{-1}$) of isotopically enriched diamond. Besides, a reference from other group (Yong Zhang et al. *Adv. Funct. Mater.* **2015**, 25, 4430–4435) reported similar data, where c of graphene-based film was $0.76 \text{ J g}^{-1} \text{ K}^{-1}$ (measured by the Hot Disk transient method), and the thermal diffusivity was $18.0 \text{ cm}^2 \text{ s}^{-1}$ (determined by the Nanoflash method).

For measurement of thermal conductivity on our GFFs, the laser flash method is not applicable due to the network structure which is permeable to incident laser. Therefore, the currently used self-heating method is an alternative and effective way to survey the thermal conductivity of GFFs. The typical measurement data for calculating the thermal conductivity of GFFs have been provided in Supplementary Figure 4b as the reviewer suggested.

Since the laser flash measurement could be performed on nonporous graphene films, further proof acquired from laser flash analysis showing the extremely high thermal diffusivity of our

compact graphene films is included in Review Only Materials.

For Reviewer #3:

Comment: Reviewer #1 raised two remarks. I will give my personal opinions on each of them, as follows.

(1) “The results reported are not state-of-the-art as compared with the current fields....”

This statement is quite subjective. However, I assume the reviewer #1 looks into the individual data presented, instead of the overall picture of the paper in a more holistic way.

The nonwoven GFFs developed in this work are conceptually new and indeed possess multifunctional capabilities with exceptional transport properties, permeability, flexibility and light weight, which are demonstrated to be useful as electrodes for supercapacitors and batteries, sensitive electrothermal heaters and durable oil-adsorbing felts. In view of this, the paper has sufficient merits on its originality and overall quality to recommend its publication in the Journal.

(2) “The key observation of the excellent specific thermal conductivity is questionable.”

The thermal conductivity of 301.5 W/mK for the GFF with a density of 0.22 g/cm³ is indeed exceptionally high. Nevertheless, I do not find the result too surprising for the following reasons:

(i) GFF was heat-treated at a very high temperature of 3000 degree C, which greatly helped the reduction of GO. Such high-temperature treatments drastically improve the thermal conductivities of carbon materials, e.g. pitch-based carbon foam (Carbon 2000, 38, 953–973) and graphene papers (Adv. Funct. Mater. 2014, 24, 4542–4548), because of the restoration of the graphitic structure. Such a high degree of graphitization is supported by the extremely high electrical conductivity of 280 S/cm and the Raman spectra.

(ii) The interfused junction can significantly reduce the interface resistance, which also greatly improve the thermal conductivity.

(iii) The fibers were aligned in the plane along which the thermal conductivity was measured. This can also lead to a high thermal conductivity.

However, I also agree with Reviewer #1 in that a higher density can result in a higher thermal conductivity, especially for the porous materials (see Carbon 2000, 38, 953–973; Nano Lett. 2012, 12, 2959–2964). One possible way for the authors to further confirm the validity of the result is to

measure the thermal conductivities of GFFs with different packing densities of graphene fibers so that the thermal conductivities indeed increase systematically with the density of GFF. The original density of carbon, 2.2 g/cm^3 , may not be achieved, but at least a trend or correlation can be established which is informative for the validity of the thermal conductivity results.

Response: Thanks a lot for the kind support and advice. We measured the electrical and thermal conductivities of 130GFFs with different packing densities as the reviewer suggested (Supplementary Figure 4a) (see below). In the density range under investigation, the electrical and thermal conductivities both increase systematically with the density of GFFs indeed, which is in accordance with the published references and the prediction of Reviewer #3.